# ASTRA🌍: GENERAL INTERACTIVE WORLD MODEL WITH AUTOREGRESSIVE DENOISING

**Yixuan Zhu**[1,*], **Jiaqi Feng**[1,*], **Wenzhao Zheng**[1,†],
**Yuan Gao**[2], **Xin Tao**[2], **Pengfei Wan**[2], **Jie Zhou**[1], **Jiwen Lu**[1,✉]
[1] Tsinghua University   [2] Kuaishou Technology

Project Page: https://eternalevan.github.io/Astra-project/
Code: https://github.com/EternalEvan/Astra

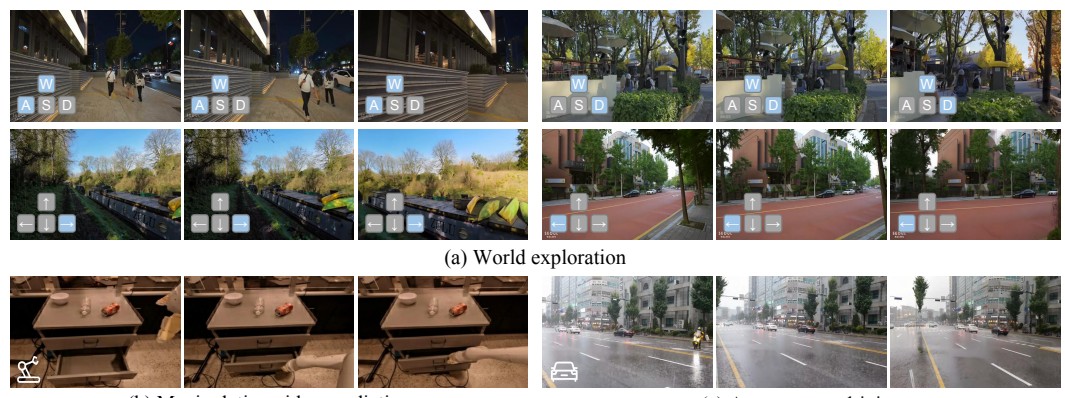

(a) World exploration

(b) Manipulation video prediction          (c) Autonomous driving

Figure 1: Our *Astra* enables interactive and versatile world modeling across exploration, robotics, and autonomous driving. With our enhanced design spanning framework architecture to training and inference, it delivers precise responsiveness to user instructions and strong long-term consistency, achieving coherent high-fidelity videos that faithfully follow instructions.

## ABSTRACT

Recent advances in diffusion transformers have empowered video generation models to generate high-quality video clips from texts or images. However, world models with the ability to predict long-horizon futures from past observations and actions remain underexplored, especially for general-purpose scenarios and various forms of actions. To bridge this gap, we introduce Astra, an interactive general world model that generates real-world futures for diverse scenarios (e.g., autonomous driving, robot grasping) with precise action interactions (e.g., camera motion, robot action). We propose an autoregressive denoising architecture and use temporal causal attention to aggregate past observations and support streaming outputs. We use a noise-augmented history memory to avoid over-reliance on past frames to balance responsiveness with temporal coherence. For precise action control, we introduce an action-aware adapter that directly injects action signals into the denoising process. We further develop a mixture of action experts that dynamically route heterogeneous action modalities, enhancing versatility across diverse real-world tasks such as exploration, manipulation, and camera control. Astra achieves interactive, consistent, and general long-term video prediction and supports various forms of interactions. Experiments across multiple datasets demonstrate the improvements of Astra in fidelity, long-range prediction, and action alignment over existing state-of-the-art world models.

---

*Equal contribution; †Project leader; ✉Corresponding author.

# 1 INTRODUCTION

Building generative world models is an emerging field where the ability to synthesize realistic and coherent video trajectories serves as a proxy for understanding and simulating the underlying dynamics of the world. With the rapid advances in visual generation (Rombach et al., 2022; Blattmann et al., 2023; Yang et al., 2025; Brooks et al., 2024; Wan et al., 2025), numerous video generation models have emerged and can perceive contextual cues and synthesize high-fidelity videos of open-world scenarios. These advances serve as the foundation for broader world simulation tasks, including game engines, autonomous driving, and spatial intelligence.

Standard text-to-video (T2V) or image-to-video (I2V) models typically produce only short, self-contained video clips conditioned on prompts or reference images. They lack the ability to generate coherent long-horizon rollouts that respond adaptively to external stimuli such as agent movements, viewpoint changes, or control signals. Without this responsiveness, these models fall short of simulating the interactive and causal dynamics of the real world. Furthermore, existing video generators are constrained by the finite temporal window of diffusion models, which prevents them from producing extended video sequences. Although recent works (Mao et al., 2025; He et al., 2025; Song et al., 2025; Teng et al., 2025) have explored video continuation techniques or hybrid frameworks that combine autoregression with diffusion, they often struggle to strike a balance between maintaining consistency with historical frames and remaining responsive to new inputs. In addition, the autoregressive generation process introduces error accumulation, leading to degraded quality and coherence in long-term predictions. As a result, despite impressive progress in generative fidelity, existing approaches remain largely passive—capable of rendering visually compelling content yet lacking the interactivity, adaptability, and robustness required for true world simulation.

To address these challenges, we present Astra, a simple yet powerful framework for building highly interactive world models. At its core, our method adopts an autoregressive denoising paradigm, where we augment a pre-trained video diffusion backbone with an action-aware adapter, as shown in Figure 2. This design preserves the high generative quality of diffusion models while enabling precise conditioning on agent actions, thereby allowing the model to produce coherent futures that respond instantly to user inputs. A central difficulty in world modeling is balancing long-term temporal consistency with action responsiveness. To mitigate this, we propose a noise-as-mask strategy that softly corrupts historical frames during training. This reduces the dominance of visual context, forcing the model to integrate both history and action cues when predicting the next video chunk. Moreover, real-world interactive environments involve heterogeneous action modalities—from camera controls and body pose to robot manipulations. To enhance versatility across such settings, we design a Mixture of Action Experts (MoAE), where modality-specific experts specialize in different action types under a learnable routing mechanism. This enables our model to unify diverse interaction signals within a single framework, making it broadly applicable across scenarios spanning embodied robotics, immersive video simulation, and long-horizon world exploration.

We conduct extensive experiments on diverse open-source benchmarks, including Sekai (Li et al., 2025a), SpatialVID (Wang et al., 2025), RT-1 (Brohan et al., 2022), nuScenes (Caesar et al., 2020) and Multi-Cam Video (Bai et al., 2025). As illustrated in Figure 1, Astra achieves state-of-the-art performance in action-driven video prediction, generating sequences that are highly interactive while maintaining visual coherence and dynamic consistency. Furthermore, our framework also demonstrates strong generalization across tasks and environments, underscoring its potential as a foundation for next-generation visual world models.

# 2 RELATED WORK

**Video generation models**. In recent years, denoising diffusion models (Ho et al., 2020; Song et al., 2020) have become the dominant paradigm in generative modeling, celebrated for their high fidelity and controllability. Following their success in text-to-image (T2I) synthesis (Rombach et al., 2022; Dhariwal & Nichol, 2021), early video generation methods (Blattmann et al., 2023; Ho et al., 2022) extended diffusion models to the temporal domain, typically by inflating image-based UNets (Ronneberger et al., 2015) with additional temporal layers. More recently, Sora (Brooks et al., 2024) and following studies (Zheng et al., 2024; Yang et al., 2025; Ma et al., 2024) pioneer high-quality, high-resolution text-to-video (T2V) diffusion models. These models commonly use the diffusion

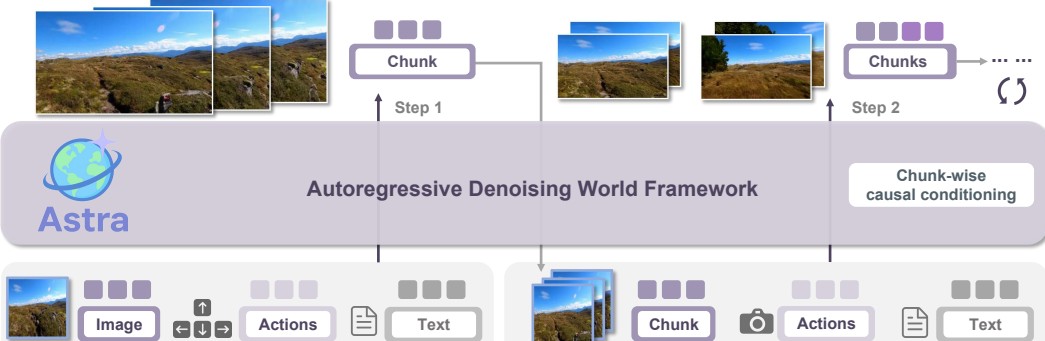

Figure 2: **Overview of the proposed *Astra*.** Our autoregressive denoising world model generates future video chunk by chunk from an initial image, actions, and optional prompts. Chunk-wise causal conditioning enforces temporal coherence and faithful action response.

transformer (DiT) architecture (Peebles & Xie, 2023) to capture complex spatial and temporal coherence. More recent models, including (Song et al., 2025) and (Zhang & Agrawala, 2025), further improve long-horizon consistency. Beyond pure diffusion, hybrid frameworks have been proposed to reconcile long-range prediction with high-quality synthesis. By combining autoregression for temporal modeling and diffusion for local realism, approaches such as StreamingT2V (Henschel et al., 2025) and MAGI (Teng et al., 2025) achieve extended video rollouts. However, these methods still struggle with error accumulation across long sequences and offer limited responsiveness to external actions, leaving a gap between video generation and true interactive world simulation.

**Visual world models**. Beyond video generation, visual world models aim to capture the causal dynamics of the environment, enabling agents to simulate future trajectories and interact with the world. Unlike text-to-video or image-to-video models that generate short clips conditioned on static inputs, visual world models explicitly incorporate history and actions, making them essential for tasks such as planning, control, and embodied intelligence. Several recent works demonstrate this trend. (Wu et al., 2024) extends autoregressive video transformers to integrate actions and rewards, allowing agents to predict how future observations evolve in interactive environments. (Wang et al., 2024) formulates world modeling as masked-token prediction in discrete latent space, supporting multimodal conditioning and open-ended environment simulation. (Huang et al., 2025; He et al., 2025) adapts pretrained video diffusion models into an autoregressive framework with causal action guidance, making them capable of controllable video prediction. In navigation contexts, (Bar et al., 2025) employs conditional diffusion-transformers to generate plausible future agent observations, facilitating planning in unfamiliar scenes. Meanwhile, (Cen et al., 2025) introduces a joint vision-language-action (VLA) framework that learns to model both world states and agent behaviors in an autoregressive manner. At larger scales, frameworks such as (Bruce et al., 2024; Agarwal et al., 2025; Liu et al., 2024) demonstrate that scaling up transformer architectures and extending context windows significantly improve rollout fidelity and generalization across domains. Recently, (Mao et al., 2025) uses a Masked Video Diffusion Transformer (MVDT) that masks input features to improve video quality. Astra instead adopts a noise-as-mask strategy that selectively degrades past visual context, prompting the model to better integrate action signals and follow user commands. While such approaches demonstrate the promise of world models for interactive environments, they still face error accumulation, temporal drift in long rollouts, and limited responsiveness to diverse actions. Overcoming these issues remains key to building reliable and scalable visual world models.

## 3 PROPOSED APPROACH

In this section, we present Astra, an autoregressive denoising framework that achieves real-world video prediction and enjoys high interactivity, versatility, and consistency. Our key idea is to harness the visual generation power within a pre-trained text-to-video diffusion model and incorporate the chunk-wise autoregressive prediction by using previously generated clips as conditions. We will start by reviewing the background of autoregressive denoising models, and then describe our designs of Astra, including an action-aware adapter for precise conditioning and a noise-augmented history memory for long-term consistency, and a mixture of action experts (MoAE) for unifying diverse action inputs. The overall framework of our Astra is illustrated in Figure 3.

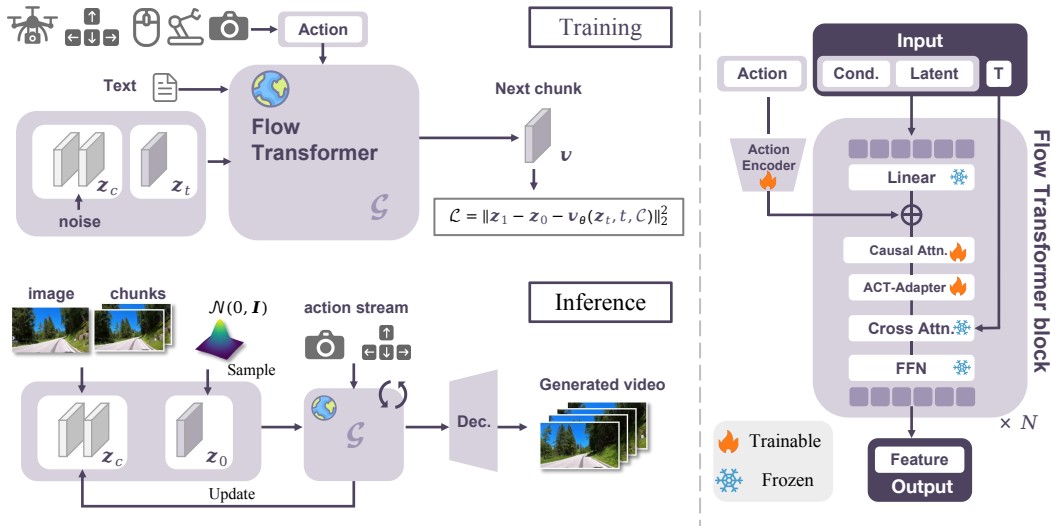

*(a) Autoregressive denoising world framework*          *(b) Action-aware flow transformer*

**Figure 3:** **The overall framework of *Astra*.** The Action-Aware Flow Transformer (AFT) injects action signals into the latent space via an ACT-Adapter (right), which aligns action features through an encoder and adds them to each transformer block. During training (left top), the model learns next-chunk prediction with flow matching. During inference (left bottom), it autoregressively generates video chunks conditioned on history and action streams, producing interactive videos.

## 3.1 PRELIMINARY: AUTOREGRESSIVE DENOISING MODEL

We adopt an autoregressive denoising framework, which integrates the long-horizon modeling of autoregression with the high-fidelity synthesis of diffusion. Given a video sequence discretized into chunks $z^{1:N}$, the generation objective is:

$$p(\boldsymbol{z}^{1:N}) = \prod_{i=1}^{N} p(\boldsymbol{z}^i \mid \boldsymbol{z}^{<i}). \tag{1}$$

For each step, the next chunk $\boldsymbol{z}_{t+1}$ is predicted through a denoising process, trained using flow matching. Specifically, we first sample a noisy interpolation of the target chunk:

$$\boldsymbol{z}_t^i = (1-t)\boldsymbol{z}_0^i + t\,\boldsymbol{\epsilon}, \quad \boldsymbol{\epsilon} \sim \mathcal{N}(0, I),\ t \in [0,1], \tag{2}$$

and train the flow model $\boldsymbol{v}_\theta$ to estimate the clean direction:

$$\mathcal{L}(\theta) = \mathbb{E}_{i,t,\boldsymbol{\epsilon}}\Big[\|\boldsymbol{v}_\theta(\boldsymbol{z}_t^i, t \mid \boldsymbol{z}^{<i}) - \boldsymbol{v}^*(\boldsymbol{z}_t^i, t \mid \boldsymbol{z}^{<i})\|_2^2\Big], \tag{3}$$

where $\boldsymbol{v}^*$ is the ground-truth velocity. In inference, autoregressive generation proceeds by denoising from noise to obtain $\boldsymbol{z}^{i+1}$, which is then appended to history for predicting future chunks. This iterative AR–denoising loop enables long-range, consistent, and high-quality video prediction.

## 3.2 INTERACTIVE WORLD MODELING VIA AUTOREGRESSIVE DENOISING

Modern video generation models (Brooks et al., 2024; Wan et al., 2025) have demonstrated remarkable progress in simulating realistic visual dynamics. These models benefit from large-scale pretraining, which enables them to implicitly acquire partial knowledge of 3D spatial perception, temporal dependencies, and even simple physical patterns such as motion and force. However, despite their impressive fidelity, these models still fall short in constructing real-world scenes that can be preserved, interacted and explored. This raises a key question: *Are T2V models really world models?* In our view, the defining property of a world model is interactivity—the ability to adapt generation dynamically in response to arbitrary action inputs at arbitrary moments. While diffusion-based models can be conditioned on global prompts or scene attributes, such conditioning mechanisms do not enable fine-grained, online interaction. To address this limitation, we turn to the autoregressive framework, which naturally supports stepwise prediction conditioned on both past observations

and current actions. Unlike diffusion models that generate video in a single pass, autoregression allows for instantaneous responses to action inputs, enabling controllable and adaptive rollouts. By integrating this property with the generative power of denoising models, we design an autoregressive denoising framework that achieves both high-quality synthesis and interactive controllability. Although prior works have explored combining autoregression and denoising, reconstructing this hybrid paradigm for world modeling remains non-trivial. Beyond simply chaining autoregression and denoising, we must carefully define the observation–action interface and design mechanisms to balance the trade-off between long-term consistency and immediate responsiveness

Given the previous video chunks $z^{1:i-1}$, we aim to model the conditional distribution of the next chunk $p(z^i|z^{1:i-1})$. In principle, this prediction can be realized by a variety of generative models. To ensure high visual fidelity, we choose to leverage a pre-trained video flow matching model $v_\theta$ as the predictor, leveraging its strong video synthesis capability. However, integrating such a model into an interactive world framework introduces two central challenges: (1) How to represent actions and quantify their impact on future visual dynamics; (2) How to effectively incorporate action signals into a pre-trained diffusion backbone while preserving its generative quality.

Since our goal is to enable instantaneous responses to action inputs $a^i$ (*e.g.*, an instruction such as "turn right"), the effect of an action should manifest as a direct transformation on the predicted video chunk. Inspired by the formulation of optical flow, we interpret this transformation as a shift in video features, which, in diffusion models, correspond to latent representations within the denoiser. Accordingly, we treat the action as an additional conditioning signal for the diffusion model, applied directly to its latent feature space. This requirement poses a challenge for existing video diffusion architectures, which are typically composed of stacked Transformer blocks (DiT) and rely on cross-attention layers to align video latents with textual embeddings. Such mechanisms are not naturally suited for modeling fine-grained action-induced shifts. To overcome this, we introduce the action-aware flow transformer adapter (ACT-Adapter) that augments a pre-trained video DiT into an autoregressive denoising model capable of integrating action signals as latent-space transformations, preserving the generative power of the backbone while modeling the action's influence.

As shown in Figure 3, we introduce an action encoder to project actions into a feature space aligned with the video latents. The resulting action features are injected into the denoising model through element-wise addition at every block, ensuring that action signals directly modulate the latent representation. To maximize reuse of pre-trained knowledge, we freeze all parameters of the flow transformer except for the self-attention layers. In addition, we insert a lightweight adapter module—a single linear layer initialized as the identity matrix—after each self-attention block. This enables the model to gradually learn action-aware transformations while maintaining the stability of the pre-trained backbone. For the history condition $z_c = z^{1:i-1}$, we adopt the frame-dimension conditioning strategy that concatenates the previous chunks with the predicted chunk along the temporal dimension before being processed by the flow transformer. Together with the action $a^i$ and prompt $c$, the full condition set for the denoising model $v_\theta$ is $\mathcal{C} = \{z^{1:i-1}, a^{1:i}, c\}$.

To enhance the effect of actions, we propose an action-free guidance mechanism (AFG), inspired by class-free guidance (CFG). During training, action conditions are randomly dropped, forcing the model to predict without action inputs. At inference, we compute a guided velocity field:

$$v_{\text{guided}} = v_\theta(z_t, t, \emptyset) + s \cdot (v_\theta(z_t, t, a) - v_\theta(z_t, t, \emptyset)), \tag{4}$$

where $s$ is the guidance scale, $z_t$ is the combined latent and $\emptyset$ denotes the null-action condition. This technique sharpens the action effect, yielding more precise responses to user inputs.

### 3.3 HISTORY CONDITION WITH NOISY MEMORY

After addressing the challenge of action control, we turn to another open problem: balancing the long-term temporal consistency with responsiveness to actions. Prior works have shown that generating coherent long videos requires conditioning on an extended history. However, we observe a trade-off: increasing the length of history improves temporal consistency but weakens action responsiveness. We refer to this phenomenon as *visual inertia*—the tendency of the model to rely heavily on past visual information while overlooking user actions. This arises because real-world datasets contain predominantly smooth motions, leading the model to prioritize continuity over abrupt, action-driven changes. To mitigate this inherent contradiction, we avoid naively shortening the conditioning horizon and instead seek a more elegant solution. Considering the asymmetry

Figure 4: **Mixture of Action Experts (MoAE).** Action signals from diverse modalities are projected into a shared space, augmented with a history mask, and routed to modality-specialized experts. A dynamic router selects top-k experts, whose outputs are aggregated into unified embeddings and fed into the Flow Transformer, enabling versatile and precise action-conditioned generation.

between dense visual inputs and sparse action signals, we propose to reduce the dominance of visual conditioning by introducing controlled corruption. Unlike (Mao et al., 2025), which randomly masks visual tokens, we adopt a noise-as-mask strategy: injecting random noise into the conditioning video to degrade and blur its information content (Figure 3). This design offers two advantages. First, it requires no architectural modifications or additional learnable parameters in the denoising model. Second, by corrupting the visual context, it prevents the model from directly copying clean frames and forces it to integrate action cues into generation. The corruption noise is independent of diffusion noise, so inference can use clean historical frames. Through this training strategy, the model learns to balance reliance on action and history, thereby overcoming visual inertia. To further extend the effective history horizon, we adopt the compression approach of (Zhang & Agrawala, 2025), which retains the first frame while compressing the intermediate history into compact visual tokens, preserving long-range temporal information without overwhelming the action signal.

### 3.4 MIXURE OF ACTION EXPERTS FOR DIVERSE SCENARIOS

Interactive world modeling often involves multi-modal inputs, including camera observations, body pose, and discrete action commands. These heterogeneous signals differ in structure and scale, making it challenging for a single model to capture their characteristics. To address this, we propose the Mixture of Action Experts (MoAE), a modular framework that routes different modalities to specialized experts, producing a unified action representation for the denoising model.

As shown in Figure 4, each action modality—continuous camera pose $a_{\mathrm{cam}}$, robot pose $a_{\mathrm{rob}}$, and discrete keyboard/mouse commands $a_{\mathrm{cmd}}$—is first mapped into a shared action space via a modality-specific projector $\mathcal{R}_m$ as $\widetilde{a}^i = \mathcal{R}_m(a_m^i)$, where $m \in \{\mathrm{cam}, \mathrm{rob}, \mathrm{cmd}\}$ denotes the specific modality and $i$ is the sequence index. A router network then computes gating scores $g^i = \mathrm{Router}(\widetilde{a}^i)$ to select the top-$K$ relevant experts. Each chosen expert $E_k$, implemented as independent MLPs, transforms the aligned features into task-relevant representations. The expert outputs are then aggregated according to the router's gating scores to produce the final action embedding $e^i = \sum_{k=1}^{K} g_k^i E_k(\widetilde{a}^i)$. The embedding sequence $e^{1:i}$ is then fed into the flow transformer. To account for both historical and current actions, we augment the action space of $\widetilde{a}^i$ with an additional binary indicator specifying whether the input corresponds to a past or current action.

Combining MoAE with history-conditioned latent inputs allows the model to generate video chunks that are temporally coherent and responsive across modalities. This design improves modality specialization, scalability to new signals, efficiency by activating only relevant experts, and overall versatility, enabling high-fidelity predictions in complex interactive scenarios.

## 4 EXPERIMENT

### 4.1 EXPERIMENT SETUPS

**Datasets.** To train our model, we leverage a diverse set of datasets covering autonomous driving, egocentric exploration, multi-camera rendering, and robot control, as shown in Table 1. Specifically, we use nuScenes (Caesar et al., 2020) for vehicle pose prediction, Sekai (Li et al., 2025a) and

Table 1: Datasets used in experiments, along with their actions and sample sizes. For each dataset, we list the action type, followed by the dimensionality of its representation.

| Dataset | Action | Scenario | Size |
|---|---|---|---|
| nuScenes (Caesar et al., 2020) | Camera (7) | Autonomous driving | 850 |
| Sekai (Li et al., 2025a) | Camera (12) | Walking & drone view | 50K |
| SpatialVid (Wang et al., 2025) | Camera (7) & keyboard / mouse | In-the-wild videos | 200K |
| RT-1 (Brohan et al., 2022) | Robotic pose (7) | Robot manipulation | 9978 |
| Multi-Cam Video (Bai et al., 2025) | Camera (12) | Human motion | 136K |
| Total | - | - | $\sim$ 397K (360 hours) |

Table 2: **Quantitative comparison of different models.** Astra demonstrates superior visual quality and instruction-following performance across a variety of real-world scenarios.

| Method | Instruction Following ↑ | Subject Consistency ↑ | Background Consistency ↑ | Motion Smoothness ↑ | Aesthetic Quality ↑ | Imaging Quality ↑ |
|---|---|---|---|---|---|---|
| Wan-2.1 (Wan et al., 2025) | 0.061 | 0.854 | 0.903 | 0.958 | 0.489 | 0.691 |
| MatrixGame (He et al., 2025) | 0.268 | 0.916 | 0.928 | 0.981 | 0.441 | **0.748** |
| Yume (Mao et al., 2025) | 0.652 | 0.936 | 0.938 | 0.985 | 0.523 | 0.741 |
| Astra (Ours) | **0.669** | **0.939** | **0.945** | **0.989** | **0.531** | 0.747 |

SpatialVID (Wang et al., 2025) for large-scale in-the-wild videos with rich camera annotations, Multi-Cam Video (Bai et al., 2025) for synthetic multi-view sequences, and RT-1 (Brohan et al., 2022) (via Open X-Embodiment (O'Neill et al., 2024)) for robot action trajectories. All videos are resized and cropped to 480p, and action annotations are temporally aligned by interpolating to every 4 frames, matching the temporal compression of the video VAE. Together, these datasets provide complementary action signals (vehicle, camera, and robot pose) that support unified action-aware world modeling. For evaluation, we construct Astra-Bench, a benchmark comprising 20 held-out samples from each dataset, designed to cover a diverse range of real-world scenarios.

**Training details.** We initialize our model from the pre-trained video diffusion model (Wan et al., 2025) and train it on 8 GPUs (80 GB each) with a per-GPU batch size of 1. Optimization is performed with AdamW (Loshchilov & Hutter, 2017) using a learning rate of $1e-5$ for 30 epochs, requiring about 24 hours to converge. The training is conducted in the latent space of a 3D VAE. In pixel space, the number of condition frames is randomly sampled from [1, 128], while the number of target frames is fixed to 33. Additional implementation details are provided in Section A.

**Metrics.** Astra-Bench evaluates two key aspects of world models: visual quality and instruction following (camera motion tracking), using six fine-grained metrics. For instruction following, we assess whether generated videos accurately reflect intended walking directions and camera movements. While pose estimation tools such as MegaSaM (Li et al., 2025b) can automate this process, inaccuracies in camera motion prediction and quantization errors limit their reliability. We therefore adopt human evaluation to ensure accurate assessment. We adopt metrics from VBench (Huang et al., 2024) for the remaining dimensions (i.e., subject consistency, background consistency, motion smoothness, aesthetic quality, and image fidelity). All test videos are generated at 480×832 resolution, 20 FPS, and 96 frames, using 50 inference steps for each model.

## 4.2 MAIN RESULTS

We evaluate our method on Astra-Bench and compare it against recent state-of-the-art video generation and world modeling approaches, including Wan-2.1 (Wan et al., 2025), Matrix-Game (He et al., 2025) and YUME (Mao et al., 2025). As shown in Table 2, our method consistently outperforms baselines across all metrics, demonstrating strong advantages in both visual quality and action-conditioned responsiveness. Astra achieves consistently superior results, surpassing existing video generation and world modeling approaches in both fidelity and controllability. Our model produces videos with sharper details, smoother motion, and stronger temporal coherence (Figure 5), leading to higher scores on visual quality metrics such as subject and background consistency, motion smoothness, and overall aesthetic appeal. More importantly, it demonstrates a clear advantage in instruction following: human evaluations confirm that the generated trajectories more faithfully follow intended camera movements and action directions compared to prior methods. Importantly, while competing approaches often suffer from error accumulation and drift during long rollouts,

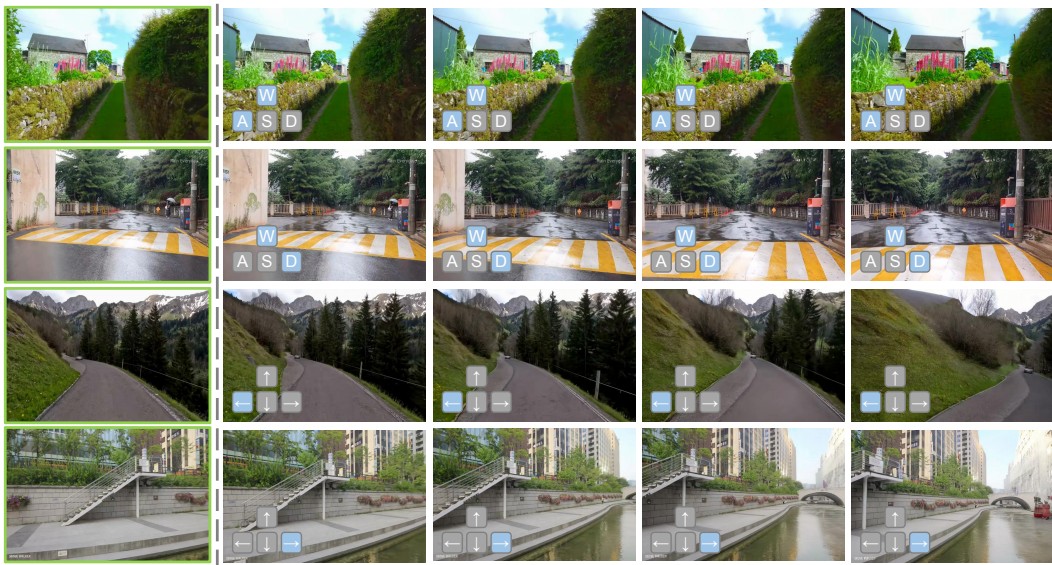

Input image            Action-driven video generation

Figure 5: **Qualitative results on action-driven real-world exploration.** Starting from a single initial frame, our model generates long-term exploration videos with high visual fidelity, smooth and coherent dynamics, and precise responsiveness to action inputs.

Astra maintains stability across extended horizons (Figure 6), making it particularly suitable for real-world interactive scenarios where both high fidelity and reliable action following are critical.

## 4.3 ANALYSIS

**Action integration via ACT-Adapter.** Our lightweight ACT-Adapter provides an efficient way to inject action features into the pre-trained flow transformer. Freezing most parameters while tuning only the adapter and attention layers ensures maximum reuse of generative knowledge. Ablation results (Table 3, cross attn. adapter) demonstrate that ACT-Adapter achieves stronger action-conditioned performance than the cross-attention adapter used in He et al. (2025), confirming its effectiveness as a simple yet powerful extension for enhancing interactivity.

**Action-free guidance enhances action responsiveness.** We find that action-free guidance effectively amplifies the influence of action signals during inference. By learning to predict both with and without actions, the model gains stronger controllability, achieving sharper responses to commands while preserving stability. As shown in Table 3 (w/o AFG), this mechanism proves particularly useful for improving action responsiveness in long video rollouts.

**Noisy memory mitigates visual inertia.** The proposed noise-as-mask strategy alleviates the issue of visual inertia by weakening the dominance of historical context over action inputs. This encourages the model to rely on external controls rather than simply extrapolating from past frames. As shown in Table 3 (w/o noise), the model achieves stronger responsiveness to abrupt or unexpected actions, while still maintaining long-term temporal coherence and stability in video generation.

**Adapting to diverse scenarios with MoAE.** MoAE allows Astra to process diverse action modalities in a unified way. By routing inputs to modality-specialized experts, it provides both versatility and precision across domains such as robotics and navigation. While joint training on heterogeneous datasets may slightly reduce performance on any single scenario, MoAE greatly improves cross-domain generalization and enables broader data usage—critical for real-world applications. Our ablation in Table 3 (w/o MoAE), trained only on camera-action data since it cannot process other modalities, further shows that MoAE markedly improves action-conditioned video generation.

## 4.4 EXTENDED APPLICATIONS

Our interactive world model is not limited to standard video prediction benchmarks, and it naturally extends to a wide range of real-world applications. Thanks to its balanced design of action-aware

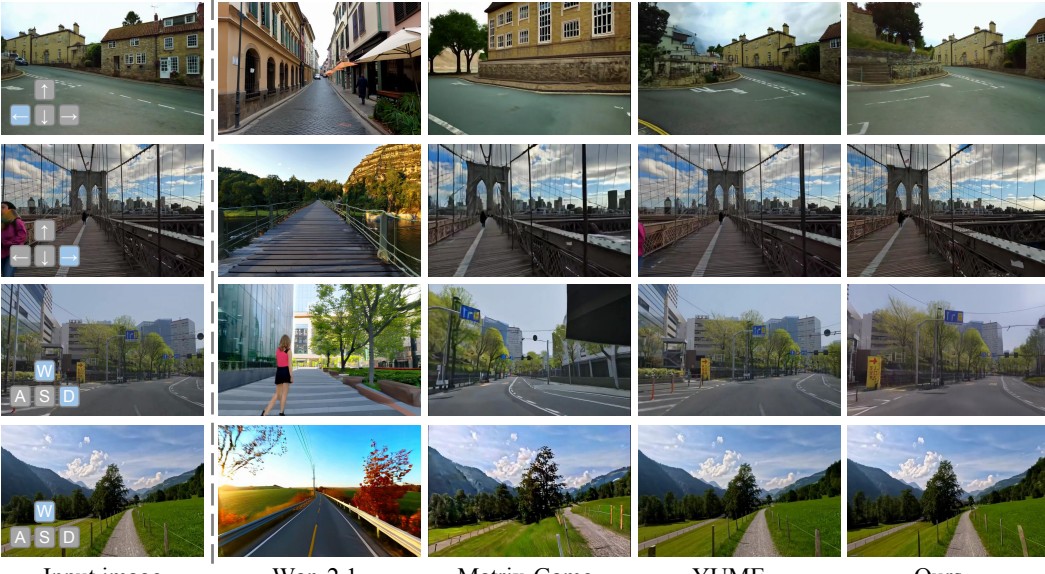

| Input image | Wan-2.1 | Matrix-Game | YUME | Ours |

Figure 6: **Qualitative comparisons on action-driven real-world exploration.** Given the initial image and action sequence, Astra generates exploration sequences that maintain strong visual fidelity, coherent dynamics, and accurate responsiveness to user-specified actions.

Table 3: **Ablation studies**. We assess the contribution of each component in Astra, ensuring all experiments are conducted using the same random seed for fair comparison.

| Method | Instruction Following ↑ | Subject Consistency ↑ | Background Consistency ↑ | Motion Smoothness ↑ | Aesthetic Quality ↑ | Imaging Quality ↑ |
|---|---|---|---|---|---|---|
| w/o AFG | 0.545 | 0.841 | 0.892 | 0.957 | 0.492 | 0.703 |
| w/o noise | 0.359 | 0.903 | 0.927 | 0.979 | 0.523 | 0.739 |
| cross attn. adapter | 0.642 | 0.926 | 0.903 | 0.948 | 0.512 | 0.694 |
| w/o MoAE | 0.651 | 0.930 | 0.941 | 0.975 | 0.520 | 0.727 |
| Astra (Ours) | **0.669** | **0.939** | **0.945** | **0.989** | **0.531** | **0.747** |

conditioning, noise-augmented memory, and modular action encoding, the model can flexibly adapt to diverse tasks such as camera control, manipulation video prediction, long-horizon exploration, and autonomous driving, as shown in Figure 7. This versatility arises from unifying temporal consistency with responsive action integration, making it a general-purpose framework for simulating, interacting with, and editing dynamic environments.

**Autonomous driving.** We extend Astra to autonomous driving using the nuScenes dataset (Caesar et al., 2020), which provides multi-view videos and diverse traffic contexts. Given ego-vehicle observations and discrete control actions (e.g., turn left, move forward), Astra generates realistic future driving videos that capture vehicle motion, road geometry, and agent interactions. Its combination of long-term coherence and precise action responsiveness enables interactive driving simulation.

**Manipulation video prediction** In robotic settings, Astra predicts future video chunks conditioned on current observations and manipulation actions, simulating fine-grained interactions such as grasping or tool use with high fidelity and temporal coherence. These predictive rollouts support planning, policy learning, and safe exploration in robot learning.

**Camera control.** Astra supports interactive camera control through action signals specifying camera trajectories such as panning and viewpoint shifts. Conditioned on these poses, it generates videos that follow instructed motions while maintaining spatial and temporal consistency. This enables controllable, cinematic camera movement for both creative and practical applications.

**Multi-agent interaction.** In Figure 8, we illustrate the ability to handle multi-agent scenarios through a first-person driving example where the ego-vehicle overtakes two cars. Conditioned on a sequence of action commands, the model produces a coherent rollout that follows the specified trajectory while plausibly simulating the motions of surrounding vehicles. This highlights that the

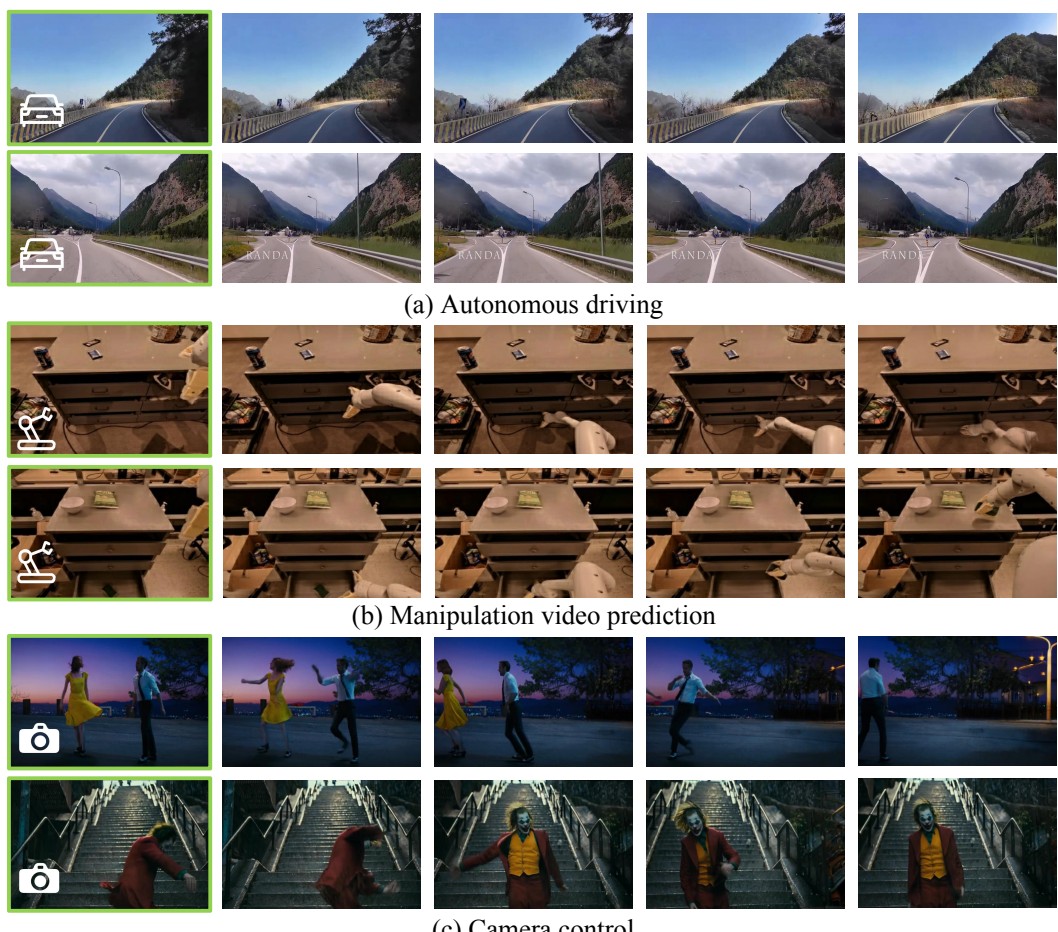

(a) Autonomous driving

(b) Manipulation video prediction

(c) Camera control

Figure 7: **Extended applications of Astra.** Our framework handles diverse scenarios: (a) autonomous driving, predicting long-horizon traffic dynamics from control inputs; (b) manipulation, conditioning robot actions on object interactions; and (c) camera control, reflecting viewpoint changes in coherent videos. These demonstrate Astra's versatility for interactive world modeling.

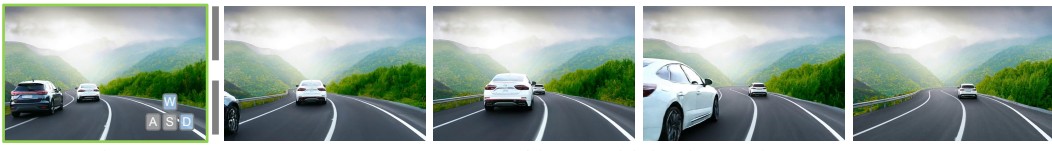

Input image                    Multi-agent driving interaction

Figure 8: **Multi-agent interaction of Astra.** Given a specified action sequence, Astra generates smooth, realistic multi-agent interactions, such as an ego-vehicle overtaking other cars.

autoregressive denoising process, noise-augmented memory, and MoAE-based action conditioning together enable stable multi-agent interactions with accurate action following.

## 5 CONCLUSION

In this work, we present Astra, a simple yet effective framework for building interactive world models that unify real-world video prediction with precise action conditioning. By equipping a pre-trained video diffusion backbone with a lightweight action-aware adapter, a noise-augmented memory for balancing history and responsiveness, and a mixture of action experts for versatile control, our model achieves interactive, consistent, and versatile video generation across diverse real-world scenarios. Extensive experiments show long-term consistency, visual fidelity, and instruction following. We believe Astra offers a practical path toward more general, scalable simulators for world modeling in exploration, robotics, autonomous driving, and embodied intelligence.

ACKNOWLEDGMENTS

This work was supported in part by the National Natural Science Foundation of China under Grant 62125603, Grant 62336004, Grant 62321005, and in part by the Beijing Natural Science Foundation under Grant No. L247009.

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

## A    MORE EXPERIMENTAL DETAILS

### A.1    DATASETS

We use the training data sampled from the nuScenes (Caesar et al., 2020), Sekai (Li et al., 2025a), SpatialVID (Wang et al., 2025), Multi-Cam Video (Bai et al., 2025) and RT-1 (Brohan et al., 2022). To ensure balanced exposure across these datasets, we apply a set of sampling weights that control the frequency with which data from each source is drawn during training. A summary of dataset statistics is provided in Table 1, with detailed descriptions reported as follows:

**nuScenes**: nuScenes is an autonomous vehicle dataset collected from an AV approved for testing on public roads and it contains the full 360° sensor suite (lidar, images, and radar). It comprises 1000 scenes, each 20s long and fully annotated with 3D bounding boxes for 23 classes and 8 attributes. We use its 7 dimensions pose parts as our action.

**Sekai**: Sekai is a high-quality first-person view worldwide video dataset with rich annotations for world exploration. It consists of over 5,000 hours of walking or drone view (FPV and UVA) videos from over 100 countries and regions across 750 cities. The action of this dataset is the camera pose.

**SpatialVID**: SpatialVID is a dataset consisting of a large corpus of in-the-wild videos with diverse scenes, camera movements and dense 3D annotations such as per-frame camera poses, depth, and motion instructions. SpatialVID has a total of 7,089 hours of dynamic content. We use the 7-dimensional camera parts as our action.

**RT-1**: RT-1 is a large, diverse dataset of robot trajectories that includes multiple tasks, objects and environments. It contains over 130k individual demonstrations constituting over 700 distinct task instructions using a large variety of objects. We use its 7 dimensions ee-space pose parts as our action. We use Open X-Embodiment (O'Neill et al., 2024) version of the RT-1 dataset.

**Multi-Cam Video**: Multi-Cam Video is a synthetic dataset rendered with multiple cameras capturing the same scene simultaneously. Animated characters are placed in diverse 3D environments, while cameras follow predefined trajectories to simulate synchronized shooting. We use 10K videos with detailed camera annotations as action signals.

### A.2    TRAINING DETAILS

We initialize our model using the pre-trained weights of the Wan-2.1 base model (Wan et al., 2025). Our training is run on 8 GPUs (80G) with a per-GPU batch size of 1. We train our model with AdamW (Loshchilov & Hutter, 2017) optimizer and a learning rate of $1e{-}5$ for 30 epochs. Our training runs take approximately 24 hours to converge. All videos are resized and cropped to the training resolution ($480 \times 832$). The model is trained on the latent space produced by a 3D VAE. In pixel space, the count of condition frames is randomly sampled from the range [1, 128], whereas the number of target frames is consistently set to 33.

### A.3    MODEL ARCHITECTURE

Our framework builds on Wan-2.1 (Wan et al., 2025), a large-scale video diffusion model composed of 30 stacked flow transformer (DiT-style) blocks, each containing multi-head self-attention and feed-forward layers with residual connections. Wan-2.1 serves as a strong pre-trained backbone, providing rich generative priors for high-quality video synthesis. To enable interactive control, we introduce two lightweight yet effective extensions: ACT-Adapter and Mixture of Action Experts (MoAE). The ACT-Adapter is implemented as a single linear layer inserted after each self-attention block. It is initialized as an identity mapping and fine-tuned jointly with the attention parameters, allowing the model to inject action features into the latent space while preserving stability of the pre-trained weights. In parallel, MoAE provides a modular action encoding mechanism. It consists of a linear router that projects heterogeneous action modalities into a shared space, followed by a set of MLP-based experts specialized for different action types. Specifically, we support camera control actions represented as 7 or 12-dimensional vectors, robotic actions represented as 7-dimensional vectors, and navigation commands expressed through discrete keyboard and mouse inputs. The routed outputs are combined into a unified representation that is fed into the ACT-Adapter, ensuring both precision and versatility. Together, these components augment Wan-2.1 into an autoregressive

Table A: **Quantitative action-alignment comparison.** We complement the human-rated instruction-following metric by reporting rotation and translation errors that directly measure how well generated camera motions align with the commanded actions.

| Method | RotErr ↓ | TransErr ↓ | Instruction Following ↑ | Imaging Quality ↑ |
|---|---|---|---|---|
| Wan-2.1 (Wan et al., 2025) | 2.96 | 7.37 | 0.061 | 0.691 |
| YUME (Mao et al., 2025) | 2.20 | 5.80 | 0.268 | 0.741 |
| MatrixGame (He et al., 2025) | 2.25 | 5.63 | 0.652 | **0.748** |
| NWM (Bar et al., 2025) | 2.47 | 6.13 | 0.311 | 0.635 |
| Astra (ours) | **1.23** | **4.86** | **0.669** | 0.747 |

denoising model capable of faithfully following actions while maintaining long-horizon temporal coherence. During inference, we set the scale $s$ for action-free guidance (AFG) to be 3.0.

### A.4 METRIC DETAILS

We follow YUME (Mao et al., 2025) to use human evaluation and VBench (Huang et al., 2024) metrics to assess the performance of our model. For instruction following, automated camera-motion estimators (e.g., MegaSaM (Li et al., 2025b)) can provide approximate motion labels, but their predictions often suffer from inaccuracies and quantization errors, especially under complex scene geometry or fast motion. To ensure reliable assessment, we therefore rely purely on human evaluation. Specifically, we recruit 20 users to inspect each generated sequence together with the corresponding action command. The instruction-following score is computed as the ratio of users who agree that the generated motion faithfully reflects the specified action direction and action type. For a more objective evaluation, we follow prior works (Bai et al., 2025; He et al., 2024) and measure how closely the camera motion in the generated video matches the ground-truth trajectory, using MegaSaM to estimate camera poses. As shown in Table A, Astra achieves lower rotation and translation errors than existing methods, consistent with the human-rated instruction-following results. Together, these metrics offer a fine-grained quantitative view of action alignment, further demonstrating the precise interactive control enabled by our model.

### A.5 PARAMETER ANALYSIS

Astra is designed to be lightweight, adding far fewer parameters than prior interactive world models such as YUME (Mao et al., 2025) and MatrixGame (He et al., 2025). As shown in Table B, MatrixGame introduces heavy cross-attention modules and large action encoders, resulting in significant training and inference overhead, while YUME depends on a much larger 13B backbone. In contrast, Astra adds only two small components: ACT-Adapter (a single linear layer after each self-attention block) and MoAE (a lightweight linear router plus small MLP experts, with only one expert active per step).

These additions contribute only a minor increase in parameters, making Astra the most parameter-efficient model among the compared methods. Because the backbone remains frozen and the added modules are shallow, the computation cost is nearly unchanged, enabling fast, stable training and long-horizon rollout without heavy architectural modifications. Overall, Astra achieves strong action-conditioned performance with the lowest parameter and compute overhead in its class.

## B COMPARISON WITH EXISTING WORLD MODELS

Table C compares four representative world models—Wan-2.1 (Wan et al., 2025), MatrixGame (He et al., 2025), YUME (Mao et al., 2025), and Astra—across three key dimensions: domain, control modality, and interaction horizon. In terms of domain, Astra is designed for general-purpose scenarios, showcasing their versatility in a wide range of applications. MatrixGame, however, is tailored specifically for game-related contexts, limiting its use to game-specific environments. YUME focuses on walking-specific domains, which is a niche area compared to the general-purpose methods. For control modalities, Wan-2.1 relies solely on text input, reflecting a language-driven paradigm. MatrixGame and YUME adopt traditional keyboard and mouse controls. Astra, enabled by MoAE, supports multiple modalities—including camera input, keyboard/mouse, and robot pose—providing more flexible and intuitive interaction. Finally, regarding the interaction horizon, Wan-2.1 and Ma-

Table B: **Parameter comparison.** Astra introduces the smallest parameter overhead among all methods, adding only lightweight adapters while preserving the efficiency of the frozen backbone.

| Method | Base model | Trainable Params. | Note |
|---|---|---|---|
| NVM (Bar et al., 2025) | CDiT-XL | $\sim$ 1B | Full tuning |
| YUME (Mao et al., 2025) | Wan2.1-14B | $\sim$ 14B | Full tuning |
| MatrixGame (He et al., 2025) | Wan2.1-1.3B | $\sim$ 1.8B | Full tuning, cross-attn adapters |
| Astra (ours) | Wan2.1-1.3B | 366.8M | Tuning adapters & self-attn. |

Table C: Comparative overview of various world model methods, detailing their respective domains of application, supported control modalities, and interaction horizons.

| Method | Domain | Control | Interaction Horizon |
|---|---|---|---|
| Wan-2.1 (Wan et al., 2025) | General | Text | A few seconds |
| MatrixGame (He et al., 2025) | Game-specific | Keyboard / Mouse | A few seconds |
| YUME (Mao et al., 2025) | Walking-specific | Keyboard / Mouse | 8-10 seconds |
| Astra (Ours) | General | Camera; Keyboard / Mouse; Robot pose | 8-10 seconds |

trixGame are limited to short spans of a few seconds. YUME extends this to 8–10 seconds, and Astra matches this longer horizon through noisy memory and the input-packing technique of (Zhang & Agrawala, 2025). Combined with its multi-modal control and general-purpose domain, Astra emerges as the most comprehensive solution among the compared methods.

# C  MORE RESULTS

## C.1  EXTENDED APPLICATIONS

As shown in Figure 7, our Astra framework is designed to generalize across a wide range of interactive video prediction tasks, demonstrating strong adaptability in domains such as autonomous driving, robotic manipulation, and camera control. In driving scenarios, Astra generates realistic long-horizon rollouts that capture complex road dynamics while responding accurately to control signals like steering or acceleration. For manipulation, it predicts object interactions conditioned on robot actions, enabling fine-grained and physically consistent outcomes. In camera control, Astra follows viewpoint instructions such as panning, zooming, or rotation, while maintaining temporal and visual coherence. Together, these applications highlight Astra's versatility and effectiveness as a unified world modeling framework capable of handling heterogeneous action modalities in diverse real-world settings.

## C.2  OUT-OF-DOMAIN GENERALIZATION

We further evaluate Astra on out-of-domain (OOD) scenes—including indoor environments, stylized anime videos, and even Minecraft gameplay—none of which are present in the training distribution. Across all cases in Figure A, Astra produces coherent, action-conditioned rollouts: given camera or navigation commands, it generates futures that accurately follow the instructed motion while maintaining global structure and temporal consistency. In indoor scenes, the model handles complex layouts and viewpoint shifts; in anime clips, it remains responsive despite fast and unpredictable dynamics; and in Minecraft, it adapts to drastically different textures and rendering styles while still executing the intended camera movement. These results show that Astra's autoregressive denoising framework, noise-augmented memory, and action-aware conditioning generalize effectively under substantial distribution shift. We also present two examples applying different complex action sequences to the same scene (Rows 3 and 4 in Figure A), both of which are followed faithfully.

## C.3  COMPARISONS WITH MORE METHODS

To more comprehensively situate Astra within the broader landscape of visual world modeling, we provide extended quantitative and qualitative comparisons with additional state-of-the-art approaches. Specifically, we compare against methods such as Navigation World Models (NVM) (Bar

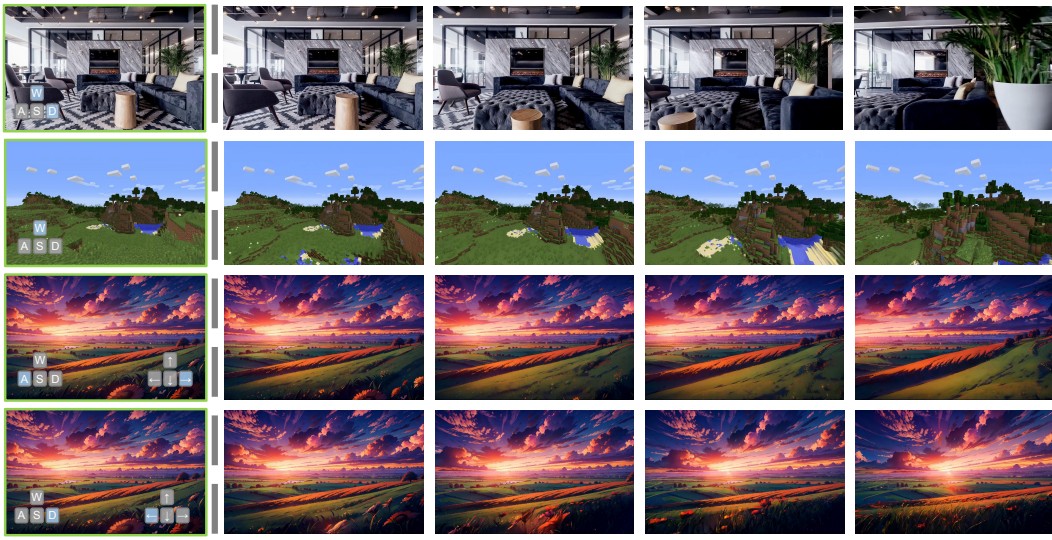

Input image · Action-driven video generation

Figure A: **Out-of-domain generation results of Astra.** Astra generalizes to scenes not seen during training, including indoor environments, Minecraft worlds, and animation-style scenes, producing coherent futures that follow camera or navigation commands. The last two rows show two distinct complex action sequences executed within the same scene.

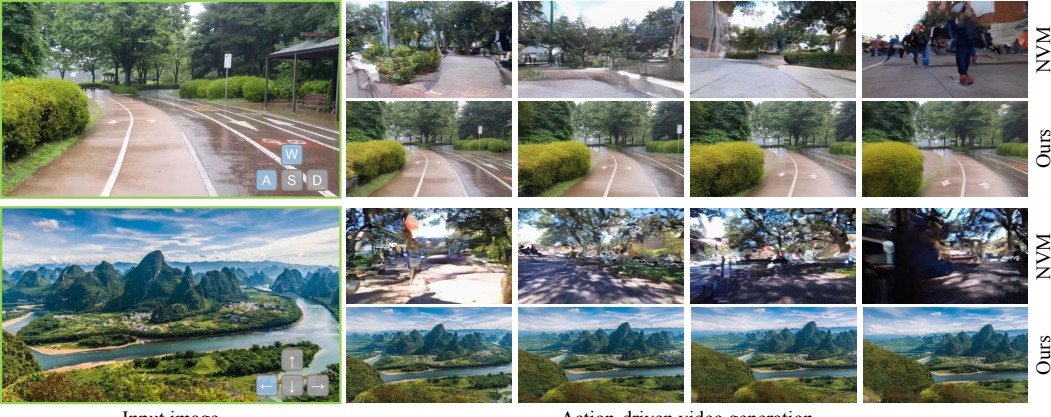

Input image · Action-driven video generation

Figure B: **Qualitative comparison with NVM** (Bar et al., 2025). Astra consistently produces visually coherent results while faithfully following action inputs.

et al., 2025) in Figure B and Table A, which are representative of recent approaches integrating multimodal inputs and action-conditioned predictions. Our results demonstrate that Astra achieves competitive or superior performance across key metrics, including visual fidelity, temporal consistency, and action responsiveness. While direct numerical comparisons are limited by differences in evaluation protocols and datasets, qualitative visualizations show that Astra generates long-horizon, coherent videos that faithfully follow user-specified actions, outperforming prior methods in capturing the causal dynamics of the environment. These comparisons highlight Astra's ability to combine accurate action conditioning with long-term rollout stability, confirming its effectiveness as a general interactive world model across diverse scenarios. It is worth noting that many of the most advanced world models, such as Genie-3 (Bruce et al., 2024), are currently closed-source or provide only limited evaluation interfaces, making direct numerical comparison infeasible.

## C.4 QUANTITATIVE COMPARISONS ON LARGER DATASETS

To further address concerns about evaluation scale, we conduct an expanded study on a larger held-out set from the CityWalker (Liu et al., 2025) dataset. CityWalker is an egocentric urban navigation corpus collected from in-the-wild YouTube walking videos with pose estimates. We sample 100

Table D: **Quantitative comparison on CityWalker dataset.** Astra consistently achieves higher visual quality and more reliable action following when evaluated on fully unseen scenes.

| Method | Instruction Following ↑ | Subject Consistency ↑ | Background Consistency ↑ | Motion Smoothness ↑ | Aesthetic Quality ↑ | Imaging Quality ↑ |
|---|---|---|---|---|---|---|
| Wan-2.1 (Wan et al., 2025) | 0.084 | 0.827 | 0.843 | 0.913 | 0.417 | 0.632 |
| MatrixGame (He et al., 2025) | 0.247 | 0.923 | 0.939 | 0.946 | 0.426 | 0.653 |
| Yume (Mao et al., 2025) | 0.619 | 0.933 | 0.927 | 0.972 | 0.511 | 0.628 |
| Astra (Ours) | **0.641** | **0.948** | **0.944** | **0.983** | **0.554** | **0.695** |

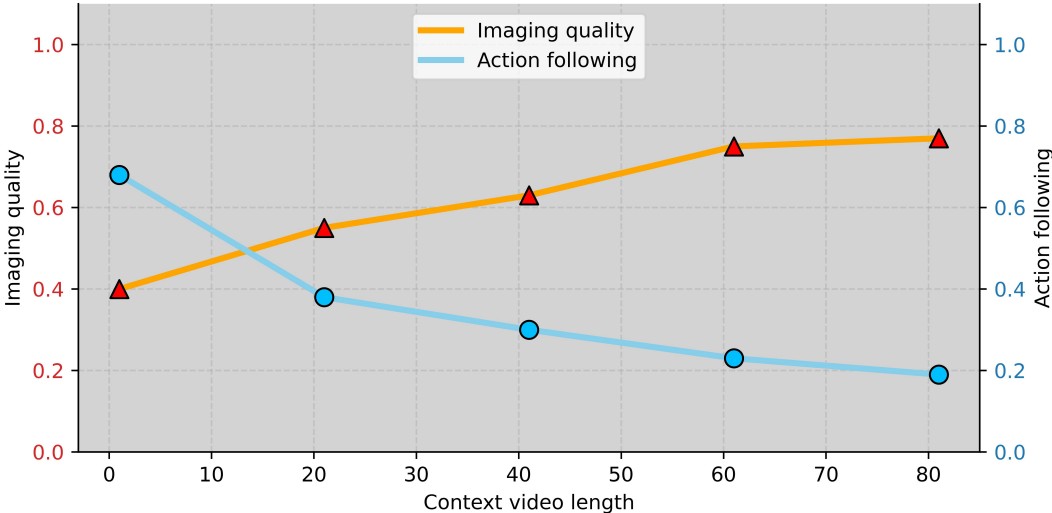

Figure C: **Effect of visual inertia.** As the history length increases, video quality improves, but the action-following score drops sharply, illustrating the visual inertia phenomenon.

scene images, each paired with its future ground-truth action trajectory, producing 100 long-horizon rollouts that robustly test cross-scene generalization and action adherence. Across this larger evaluation, Astra consistently achieves the best action-following accuracy and video quality, outperforming all baselines in metrics such as motion consistency, temporal stability, and VBench perceptual scores (Table D). These results confirm that Astra's strong performance is not an artifact of small-set evaluation, and that the model maintains robust generalization and reliable action responsiveness even under substantially expanded test conditions.

## D  DISCUSSION ON VISUAL INERTIA

Previous work (Zhang & Agrawala, 2025) has shown that longer video context can improve generation quality. However, in interactive world modeling, we observe that increasing context length can reduce action responsiveness. We term this phenomenon *visual inertia*—the model's tendency to over-rely on past visual frames while neglecting user actions. To examine this, we train Astra with different history lengths without our proposed noisy-memory mechanism. As shown in Figure C, the action-following score drops substantially as context length increases, indicating that overly clean and long visual histories can dominate the model's decision process. This motivates our noise-augmented memory design, which intentionally reduces visual dominance and encourages the model to integrate both historical context and action signals when generating future frames.

## E  LIMITATIONS

Despite the promising performance of Astra, our framework still faces limitations in inference efficiency. Since it builds on diffusion-based generation with autoregressive rollouts, producing long-horizon interactive videos requires multiple denoising steps per frame, making real-time deployment challenging. This constraint limits its applicability in latency-sensitive scenarios such as online control or interactive robotics. To address this, future work could explore distillation or student-teacher compression strategies that retain the fidelity and responsiveness of Astra while reducing inference cost, thereby paving the way for lightweight, real-time world modeling.

