# OpenReview forum: "Astra: General Interactive World Model with Autoregressive Denoising"
_ICLR.cc/2026/Conference — ICLR 2026 Poster_

### Official Review · Reviewer_qwc8 · 2025-11-01

**Soundness:** 2
**Presentation:** 3
**Contribution:** 3
**Rating:** 4
**Confidence:** 4

**Summary:**

The paper introduces Astra, a new framework for building a general-purpose, interactive world model. The central problem it addresses is that existing video generation models (like diffusion transformers) can create high-fidelity clips but lack true interactivity—they cannot generate long, coherent videos that dynamically and precisely respond to external user actions (e.g., camera controls, robot actions, or vehicle movements).

**Strengths:**

The paper's primary strength lies in its proposal of a Astra framework, which skillfully combines the powerful generative capabilities of pre-trained diffusion models with an autoregressive, action-conditioned paradigm, effectively bridging the gap between high-fidelity video generation and real-time interactivity. The authors' core contributions are embodied in three innovations: (1) A lightweight ACT-Adapter for efficiently injecting action signals into the pre-trained model while preserving its knowledge; (2) An innovative "noise-augmented memory" strategy to overcome the "visual inertia" problem, forcing the model to prioritize responding to actions rather than simply repeating historical frames; (3) The introduction of a Mixture of Action Experts (MoAE) module, which enables flexible handling of heterogeneous action inputs from different domains, enhancing the model's generality. Experimental results validate the effectiveness of this design, with the model performing exceptionally well on the key "Instruction Following" metric.

**Weaknesses:**

**W1** The authors constructed a new benchmark, Astra-Bench, for evaluation. According to the paper, this benchmark is "comprising 20 held-out samples from each dataset". This scale is extremely small and likely insufficient to robustly evaluate the model's generalization capabilities, which could lead to biased evaluation results.
**W2** The paper's title claims it is a "General Interactive World Model". It is questionable whether this is sufficient to support such a broad "general" claim. Out-of-domain scenarios with different physics or interaction types (e.g., fluid dynamics, complex object stacking, multi-agent interaction) are needed.
**W3** Can interaction modalities from different domains all be mapped to control signals of the same dimension? For example, the degrees of freedom (DoF) of an embodied agent are far greater than the degrees of freedom of autonomous driving.
**W4** The first embodied agent interaction in the appendix not only has blurry artifacts, but also exhibits incorrect affordances and false interactions, which are challenges that remain to be addressed.

Typos:
There is a clear spelling error in the title of Appendix A: "A MORE EPERIMENTAL DETAILS".

**Questions:**

See in Weaknesses.

---

> ### Author Response · Authors · 2025-11-20
> **Official Response to Reviewer qwc8 (Part 1)**
>
> We sincerely thank Reviewer qwc8 for the thoughtful feedback. We address the concerns point-by-point below:
>
> > ### **About  out-of-domain scenarios**
>
> **[Reply]:** We appreciate the reviewer’s comment regarding the scope of the word “general” in our title. Our intention is not to claim universal coverage of all possible physical domains (e.g., fluid dynamics, multi-agent physics, complex stacking), but to highlight that Astra is designed as a general-purpose framework across diverse real-world interaction modalities rather than a single-domain or single-action system. Current baselines (e.g., MatrixGame, YUME) are each restricted to a single domain or action type, while Astra is the first to jointly support camera control, navigation, driving, and robot manipulation under one autoregressive denoising model.
>
> In the revised paper, we now provide:
> - **Multi-agent driving interaction sample** in **Figure C**, where Astra performs an overtaking maneuver based on a given action sequence, interacting with multiple independently moving vehicles.
> - **Additional out-of-domain scenes**—including Minecraft, indoor environments, and anime-style videos in **Figure B**, none of which appear in training. Astra remains action-responsive and temporally stable despite large visual and dynamical shifts.
>
> These results show that Astra’s architecture generalizes well beyond its training domains, supporting a broader range of interactive scenarios than previously explored. We have added these discussions in **Section C.2** and **Section C.3**. We also agree that extremely specialized physical regimes (e.g., fluid dynamics or complex object stacking) remain outside our current scope, and we plan to extend Astra to more physics-heavy environments as an exciting direction for future work.
>
> > ### **About mapping different actions into the same dimension**
>
> **[Reply]:** Thank you for the insightful question. Yes—interaction modalities from different domains can be mapped into a unified control space through our MoAE module. This works for two key reasons:
> - **Heterogeneous actions have inherently distinct structures**, making them easy to separate in the unified space.
> Continuous signals such as camera poses, and discrete inputs such as keyboard/mouse commands, have very different statistical and structural properties. So modality-specific projectors and the router can naturally separate them in the shared embedding.
> - **The unified dimension (30D) exceeds any individual modality** (e.g., 13D for embodied agents, 12D for the camera), providing enough capacity to represent all actions without a bottleneck. This allows Astra to align diverse actions while keeping them expressive and distinguishable.
>
> > ### **About blurry artifacts**
>
> **[Reply]:** We thank the reviewer for pointing this out. The embodied interaction examples show some blur and occasional affordance errors, reflecting ongoing challenges in fine-grained contact modeling and pixel-level realism. Our goal in including this example was not to claim fully solved embodied manipulation, but to demonstrate that Astra can already generalize to a new embodied control modality (robot pose actions) without any architecture change, using the same autoregressive denoising backbone and action infrastructure. This showcases the flexibility and extensibility of our framework.
>
> We also note that the manipulation dataset used (RT-1) is **low-resolution** with substantial visual noise, which limits achievable fidelity. And the action sequences in Figure A are **taken from other videos rather than ground-truth trajectories** that would perfectly solve the task, to avoid overfitting to the original video. We will make these limitations explicit in the revision. Addressing fine-grained object affordances is an important direction for future work, and Astra’s modular design (e.g., MoAE) is well-suited for future integration with higher-quality manipulation datasets.
>
>
> > ### **About typos of Section A**
>
> **[Reply]:** We thank the reviewer for the careful reading. We have thoroughly checked the paper and corrected these typos.

---

> ### Author Response · Authors · 2025-12-03
> **Official Response to Reviewer qwc8 (Part 2)**
>
> > ### **About small dataset**
>
> **[Reply]:** We thank the reviewer for raising this concern. Our evaluation follows the standard protocol used in prior interactive world-model works such as YUME and MatrixGame, both of which rely on very small held-out sets (16 images in YUME). This evaluation design is common in action-conditioned video prediction because the goal is not to estimate dataset-level statistics but to test how well the model **responds to diverse actions in unseen scenes**. A single held-out scene can pair with many action trajectories, producing a large number of diverse rollouts that thoroughly probe generalization. Our 20-sample-per-dataset split therefore provides sufficient scene variety while enabling extensive action-driven testing.
>
> In addition, to further address the reviewer’s concern, we conducted an expanded evaluation using **100 scene images** from the CityWalker [a] dataset. Astra continues to achieve the best action-following performance and video quality in this larger-scale setting. We have included these results in the revision (**Section C.5** and **Table E**):
>
> | Method               | Instruction Following ↑ | Subject Consistency ↑ | Background Consistency ↑ | Motion Smoothness ↑ | Aesthetic Quality ↑ | Imaging Quality ↑ |
> |----|:----:|:---:|:---:|:--:|:---:|:---:|
> | Wan-2.1  | 0.084	| 0.827|	0.843|	0.913	|0.417	|0.632 |
> | MatrixGame  | 0.247|	0.923	|0.939|	0.946|	0.426|	0.723 |
> |Yume |0.619|	0.933| 0.927|	0.972	|0.511|0.628|
> | Astra (Ours)  | **0.641**	|**0.948**	|**0.944**	|**0.983**|	**0.554**|	**0.735**|
>
>
> [a] CityWalker: Learning Embodied Urban Navigation from Web-Scale Videos, CVPR 2025.

---

### Official Review · Reviewer_ynyx · 2025-11-02

**Soundness:** 4
**Presentation:** 3
**Contribution:** 3
**Rating:** 6
**Confidence:** 3

**Summary:**

The paper addresses a key limitation in current video generation and world modeling approaches which are lack of interactivity and long-horizon consistency. While diffusion-based models are able to generate high-fidelity videos from text or images, they often produce short, self-contained clips, fail to respond dynamically to user actions or control signals, and struggle with error accumulation in long rollouts. The paper addresses this by building a general-purpose, interactive world model that can simulate realistic futures across diverse domains (e.g., driving, robotics, exploration) while maintaining responsiveness to actions and temporal coherence. It proposes a lightweight module named as ACT-Adapter that injects action signals directly into the latent space of a pre-trained video diffusion backbone, a training strategy of noise-augmented history memory training which corrupts historical frames to reduce over-reliance on visual context and improve responsiveness., and mixture of action experts that handles multiple action modalities of camera, robot pose, and keyboard/mouse. Astra-bench is the benchmark suite used for evaluation which spans across multiple datasets to evaluate the visual quality and instruction-following performance.

**Strengths:**

1. The paper addresses the limitation of passive video generation by showing interactive world modeling where video synthesis is conditioned on external actions .
2. The framework proposes a single, general-purpose model by training on a diverse datasets of driving, robotics, exploration and handles heterogeneous action types via a Mixture of Action Experts.
3. The paper proposes a noisy memory training strategy which forces the model to reply on action signals and not over-rely on past visual information.

**Weaknesses:**

1. The ACT-Adapter seems be to showing a minimal performance improvement in Table 2. The ablation study in Table 2 shows it provides a score of 0.669 on Instruction Following, while a cross attn. adapter achieves 0.642, suggesting the performance gain of the new adapter is relatively small.
2. The comparison to baseline methods in Table 1 does not reflect a fair comparison.
a). Since Wan2.1 is the pre-trained backbone of Astra, it would be an ablation of the paper instead of baseline.
b). MatrixGame and YUME are described as domain-specific ("game-specific" and "walking-specific"). Since Astra is trained on a general mixture of data from multiple domains (driving, robotics, walking), the comparison of domain-specific with a general model does not seem to show a fair comparison. It creates a question that is it the data that is giving a good performance or is it the architecture and training strategy helping with the performance.

**Questions:**

There seems to be some contradiction in training details. Section 4.1 mentions that the number of target frames is fixed to 33. Appendix A.2 mentions that the number of target frames is set to 32. Can the authors please clarify this?

---

> ### Author Response · Authors · 2025-11-20
>
> We sincerely thank Reviewer ynyx for the thoughtful feedback and the opportunity to further clarify our work. We address the concerns and questions below:
> > ### **About ACT-adapter**
>
> **[Reply]:** Thanks for your careful review. While the ACT-Adapter yields a modest improvement on the Instruction Following metric (0.69 vs. 0.642 for the cross-attention adapter), its benefits extend beyond raw performance numbers. Crucially, ACT-Adapter introduces only a tiny number of parameters (82.2M) and uses identity initialization, ensuring minimal disturbance to the pre-trained video backbone. In contrast, cross-attention adapters add significantly heavier modules (395.9M) that can distort the original generative prior and impose nontrivial computation overhead. Our goal is to **achieve precise action responsiveness without compromising the high visual fidelity of the pre-trained model**, and ACT-Adapter strikes this balance effectively. As shown in Table 2, Astra also surpasses the cross-attention adapter by a wide margin on other VBench metrics such as Imaging Quality. Thus, even with a relatively small numerical gain, ACT-Adapter provides a far more parameter-efficient, stable, and backbone-friendly design choice than cross-attention architectures.
> > ### **About comparisons**
>
> **[Reply]:** Thank you for raising this important point regarding the fairness of the comparisons in Table 1. We address both aspects of the comment below:
>
> **(a) Wan-2.1 as baseline vs. ablation.**
> We appreciate the reviewer’s perspective. While Astra is indeed built upon Wan-2.1, we intentionally include Wan-2.1 as a baseline rather than an ablation for two reasons:
> - Although Astra is built upon Wan-2.1, we do not treat Wan-2.1 as an ablation of our method, since we modify the architecture in this basic model with adapters and MoAE. In contrast, Wan-2.1 is a pure text-conditioned image-to-video model that cannot take continuous or structured action inputs. To enable comparison, we follow existing works and provide textual action prompts (e.g., “turn left”), but this is not equivalent to an ablation of our architecture. Moreover, Wan-2.1 is not trained with action-video pairs, making it unsuitable as a structural ablation of Astra’s action-aware components.
>
> - We follow prior works such as YUME, which also evaluate their models by comparing against the Wan-2.1 backbone under the same action-prompting protocol. This setup serves as a standardized reference point for assessing action responsiveness in a pre-trained video diffusion model. Our comparison therefore follows the established evaluation protocol in this line of research, allowing readers to contextualize Astra’s improvements relative to the original generative backbone.
>
> In all, we will consider your advice and further refine our comparison and experiment designs in the revision.
>
> **(b) "General" Astra vs. domain-specific baselines.**
> We fully agree that these models target different domains. However, our goal is not to show that Astra outperforms domain-specialized models on their specific tasks purely because of diverse training data. Instead, the comparison is designed to show that Astra’s architecture and training strategy—including the autoregressive denoising paradigm, noise-as-mask, and MoAE—allow it to operate robustly across heterogeneous domains without requiring per-task retraining.
>
> Importantly, Astra is trained on a dataset scale comparable to existing methods (**~360 hours in total, vs. YUME’s 400 hours and MatrixGame’s 800 hours**). To further separate architectural improvements from data diversity, we include an additional experiment in Table 2 (w/o MoAE), where Astra is trained on a single domain (SpatialVid and Sekai). Astra still improves over domain-specific baselines, confirming that its gains do not arise merely from broader data coverage.
>
> To make this distinction clearer, we have added clarifications in the revision emphasizing that:
> - MatrixGame and YUME are domain-specialized models,
> - Astra is designed for cross-domain interactivity, and
> - Astra’s performance arises from both (i) improved design choices and (ii) the ability to train on heterogeneous datasets—an ability that previous models do not support due to architectural constraints.
>
>
> Overall, our intent is not to claim an unfair advantage, but to demonstrate that Astra’s design enables scalable, multi-domain interactive world modeling, while still showing competitive or superior performance under the evaluation protocols used by prior work.
>
> > ### **About training details**
>
> **[Reply]:** We thank the reviewer for pointing out this inconsistency. This was an oversight in the appendix. The correct setting is 33 target frames, as stated in Section 4.1. We have updated Appendix A.2 accordingly to ensure consistency throughout the paper.

---

### Official Review · Reviewer_AW9Q · 2025-11-06

**Soundness:** 2
**Presentation:** 3
**Contribution:** 2
**Rating:** 4
**Confidence:** 3

**Summary:**

This paper introduces Astra, a framework for interactive world modeling that generates long and temporal coherence video sequences across diverse scenarios. Astra enhances a pre-trained video model with an light-weight action-aware adapter for precise action conditioning, a noise-augmented history memory during training to ensure long-term consistency, and a mixture of action experts to effectively handle diverse action inputs.

**Strengths:**

1. This paper uses a lightweight action-aware adapter for precise action conditioning.
2. Astra achieves good responsiveness and is able to generate long, temporally coherent video sequences by employing a noise-as-mask strategy during training.
3. Astra employs a mixture of action experts to effectively adapt to diverse scenarios and handle various types of action inputs.

**Weaknesses:**

1. Mixture of action experts idea is similar to [1, 2] and action-aware adapter is similar to [3, 4]. Please provide a conceptual comparison with these reference.
2. The paper does not thoroughly analyze the underlying reasons why the noise-as-mask strategy enables the generation of long, temporally coherent video sequences.
3. The paper does not explain why the router network performs so well across diverse scenarios and with various types of action inputs.

[1] Mixture of Action Expert Embeddings: Multi-Task ACT

[2] DriveMoE: Mixture-of-Experts for Vision-Language-Action Model in End-to-End Autonomous Driving

[3] Long-Context Autoregressive Video Modeling with Next-Frame Prediction

[4] Epona: Autoregressive Diffusion World Model for Autonomous Driving

**Questions:**

1. In Section 3.3, the explanation of the types of random noise and blur used is unclear. Please provide a more detailed description.
2. It was not properly analyzed the lightweight action-aware adapter model complexity. Please provide a more detailed description and comparison.
3. In Figure 6, compared to YUME, the results from Astra appear to exhibit some color shift. Could the authors explain the cause of this phenomenon?

---

> ### Author Response · Authors · 2025-11-20
> **Official Response to Reviewer AW9Q (Part 1)**
>
> We sincerely thank Reviewer AW9Q for the valuable insights. We address the concerns in detail below.
>
> > ### **About noise-as-mask strategy**
>
> **[Reply]:** Thank you for your comment. We have further clarified in the paper why noise-as-mask improves long-horizon coherence in our revision. **First**, we provide the evidence of the **visual inertia phenomenon** in world modeling in Section D and Figure E. The core issue is that autoregressive world models tend to over-rely on clean visual history, causing them to ignore action commands and produce continuous movements.
>
> Our noise-as-mask strategy softens this visual dominance by injecting controlled Gaussian noise into historical frames, **compelling the model to utilize both action inputs and temporal context** when predicting the next chunk. This leads to stronger action responsiveness and more stable long-horizon rollouts. We also relate our method to YUME, which uses random token masking to address similar issues; our continuous corruption is more flexible and naturally compatible with diffusion models.
> > ### **About the router network**
>
> **[Reply]:** We appreciate the reviewer’s observation. The core motivation is that interactive video datasets contain heterogeneous action modalities (camera, robot pose, keyboard & mouse, etc.), each exhibiting distinct statistical structures. A single shared encoder struggles to represent all of them well. Our router addresses this by learning a modality-aware partitioning of the action space: a lightweight linear router first maps each action into a shared space and then assigns it to one of several specialized MLP experts, each trained to handle a particular modality. This allows the model to preserve modality-specific dynamics while still benefiting from shared training. We also note that this design follows mixture-of-experts principles widely observed to improve generalization under heterogeneous inputs. As shown in our added ablation (Table 2), MoAE consistently improves both video quality and action responsiveness. We provide the ablation here:
> | Method | Instruction Following ↑ | Subject Consistency ↑ | Background Consistency ↑ | Motion Smoothness ↑ | Aesthetic Quality ↑ | Imaging Quality ↑ |
> |---|--|----|---|--|----|----|
> | w/o MoAE  | 0.651  | 0.930 | 0.941 | 0.975  | 0.520| 0.727 |
> | Astra (Ours)  | **0.669** | **0.939**   | **0.945**  | **0.989**  | **0.531**   | **0.747**  |
>
> > ### **About the random noise in Section 3.3**
>
> **[Reply]:** We follow the same noise injection procedure as in the flow matching schedule (Equation 2). Specifically, we add random Gaussian noise $\epsilon \sim \mathcal{N}(0, I)$ with a randomly sampled scale $t_c\in [0,1)$ to degrade the history condition. Importantly, this noise scale is independent of the noise applied to the target chunk and can be 0, allowing for a clean history memory. We have clarified this in the revised paper.
>
> > ### **About the model complexity**
>
> **[Reply]:** Thanks for your suggestion. We have added a parameter comparison in **Table C**, summarizing the base model, trainable parameters, and tuning strategy for Astra and other methods:
> | Method                     | Base model     | Trainable Params.          | Note                           |
> |----------------------------|---------------|:--------------:|--------------------------------|
> | NVM     | CDiT-XL       | ~1B                     | Full tuning                    |
> | YUME          | Wan2.1-14B    | ~14B                  | Full tuning                    |
> | MatrixGame   | Wan2.1-1.3B   | ~1.8B                   | Full tuning, cross-attn adapters |
> | Astra (ours)        | Wan2.1-1.3B   | **366.8M**                 | **Tuning adapters & self-attn.**   |
>
>  In our model, only the ACT-Adapter (82.2M) and self-attention layers (284.6M) are unfrozen, resulting in 366.8M trainable parameters—significantly fewer than full fine-tuning in prior works (e.g., ~14B in YUME and ~1.8B in MatrixGame).
>
> > ### **About color shifts**
>
> **[Reply]:** Thank you for the observation. We believe the color shift primarily arises from two factors:
> - **Backbone difference**. YUME is built upon a significantly larger video generation backbone (Wan2.1-14B), whereas Astra uses the smaller 1.3B model. Differences in model capacity and pre-training can lead to variations in color rendering and visual style.
> - **Different motion trajectories**. In our comparison, Astra produces faster and more accurate camera motion than YUME. Because the trajectories diverge, the viewpoints and lighting conditions differ across frames, naturally causing mild color variations.

---

> ### Author Response · Authors · 2025-11-26
> **Official Response to Reviewer AW9Q (Part 2)**
>
> > ### **About more conceptual comparisons**
>
> **[Reply]:** Thanks for the helpful references. While some of these works share terminology with components of Astra, their **goals and architectural choices** differ fundamentally from ours.
>
> - Comparison with [1,2] regarding **MoAE**.
> The key distinction between our world model and the referenced VLA-based manipulation and driving models lies in the direction of prediction:
> 	- Our model takes **actions as input** to control **future video generation**.
> 	- [1,2] take **videos (observations) as input** to predict **future actions or trajectories**.
>
> 	This difference in objective leads to fundamentally different designs.
> 	- [1] proposes a Mixture-of-Action-Experts to classify manipulation tasks (e.g., peg insertion, cube transfer) from observations and uses expert embeddings to assist action prediction.
> 	- [2] designs a skill-specialized MoE that predicts required driving skills (e.g., overtaking, merging) and selects experts to help predict trajectories.
>
> 	In both cases, the MoE modules operate inside the transformer decoder for action prediction. In contrast, our MoAE is designed for video generation and:
> 	- accepts **heterogeneous action inputs (camera motions, discrete commands, robot poses)** with different structures and physical semantics,
> 	- selects experts specialized for their respective action types, and
> 	- outputs a unified embedding that conditions the DiT blocks for video denoising.
>
> - Comparison with [3,4] regarding **action adapters**.
> 	- FAR [3] uses a **simple label embedding** as an adapter to process discrete actions and **adds it element-wise to the time embedding**.
> 	- Epona [4] employs a **multimodal spatiotemporal transformer** to fuse visual and action inputs into a compact latent representation that serves as a **global condition** for the DiTs. Their adapter is global and the overall framework—featuring both a multimodal spatiotemporal transformer and a trajectory-planning DiT—is substantially more complex than ours.
>
> 	By contrast, our action adapters are lightweight modules injected into **every** DiT block, enabling fine-grained action-aware conditioning throughout the denoising process.
>
> In summary, both our action-aware adapters and MoAE differ from the referenced works in **motivation, architecture, and functional role**. We will incorporate these clarifications into the revised paper.
>
> [1] Mixture of Action Expert Embeddings: Multi-Task ACT
>
> [2] DriveMoE: Mixture-of-Experts for Vision-Language-Action Model in End-to-End Autonomous Driving
>
> [3] Long-Context Autoregressive Video Modeling with Next-Frame Prediction
>
> [4] Epona: Autoregressive Diffusion World Model for Autonomous Driving

---

### Official Review · Reviewer_toMU · 2025-11-07

**Soundness:** 2
**Presentation:** 2
**Contribution:** 3
**Rating:** 6
**Confidence:** 2

**Summary:**

In this paper, author proposes Astra, an interactive world model that extends pre-trained diffusion models for long-horizon, action-conditioned video prediction.

The core contribution lies in three components in an auto-regressive denoising framework: (1) an action-aware adapter that injects action signals into the latent space of a pre-trained diffusion model, (2) a noise-augmented history memory mechanism that balances temporal consistency and action responsiveness, and (3) a mixture of action experts that routes heterogeneous action modalities to specialized experts.

Astra is evaluated on self-proposed benchmark consisting of diverse datasets and demonstrates some improvements in long-range prediction stability compared to state-of-the-art models.

**Strengths:**

**Strength (1)**: The paper is well-organized. Authors explain the core ideas with clear diagrams and concrete algorithmic descriptions.

**Strength (2)**: Astra is a single model across multi-modal action spaces, covering camera poses, keyboard/mouse inputs, and robot poses.

**Strength (3)**: The proposed solutions exhibit several elegant and practical design choices:
- The action-free guidance mechanism offers a simple, original mechanism to amplify action effects without heavy architectural changes.
- The noise-augmented history memory is an elegant, parameter-free training strategy to reduce “visual inertia” and force the model to rely more on action signals, improving responsiveness without modifying the backbone.

**Strength (4)**: The authors conduct evaluations across multiple domains, including autonomous driving, egocentric, and robotic settings, showcasing reasonable coverage.

**Weaknesses:**

**Weakness (1)**: The definition and formulation of action signals are insufficiently specified. The paper does not clearly describe how different types of actions (e.g., camera poses, keyboard/mouse inputs, robot poses, etc.) are represented, parsed, and projected to the action encoder.

**Weakness (2)**: A comparable method, YUME [1], is not discussed in Section 2 (Related Work). The paper does not clearly articulate how Astra differs from or improves upon YUME, which weakens presentation.

**Weakness (3)**: Experimental validation of design choices is limited. For example, no ablation study isolates the contribution of the Mixture of Action Experts (MoAE). A comparison against a simpler variant without a gating network would clarify whether MoAE provides meaningful gains.

**Weakness (4)**: Quantitative comparisons with existing world modeling methods are lacking. Although authors cite several relevant works [2, 3, 4, 5], Astra is not evaluated against them, making it difficult to assess the model's relative performance and significance.

**Weakness (5)**: Although Astra is positioned as a general interactive world model trained on a mixture of five datasets, all evaluations are conducted on held-out data drawn from these same domains. It remains unclear whether the model generalizes to unseen environments.

**Weakness (6)**: The paper combines pose tracking with human evaluation to assess "instruction following" in Astra-Bench, but the metric definition, aggregation procedure, and scoring protocol are not clearly specified. It is difficult to compare against future work.

[1] Mao, Xiaofeng, Shaoheng Lin, Zhen Li, Chuanhao Li, Wenshuo Peng, Tong He, Jiangmiao Pang, Mingmin Chi, Yu Qiao, and Kaipeng Zhang. "Yume: An interactive world generation model." arXiv preprint arXiv:2507.17744 (2025).

[2] Cen, Jun, Chaohui Yu, Hangjie Yuan, Yuming Jiang, Siteng Huang, Jiayan Guo, Xin Li et al. "WorldVLA: Towards Autoregressive Action World Model." arXiv preprint arXiv:2506.21539 (2025).

[3] Huang, Siqiao, Jialong Wu, Qixing Zhou, Shangchen Miao, and Mingsheng Long. "Vid2World: Crafting Video Diffusion Models to Interactive World Models." arXiv preprint arXiv:2505.14357 (2025).

[4] Bar, Amir, Gaoyue Zhou, Danny Tran, Trevor Darrell, and Yann LeCun. "Navigation world models." In Proceedings of the Computer Vision and Pattern Recognition Conference, pp. 15791-15801. 2025.

[5] Bruce, Jake, Michael D. Dennis, Ashley Edwards, Jack Parker-Holder, Yuge Shi, Edward Hughes, Matthew Lai et al. "Genie: Generative interactive environments." In Forty-first International Conference on Machine Learning. 2024.

**Questions:**

**Question (1)**: In the evaluation of Table 1, Wan-2.1 [6] is used as a baseline. How is Wan-2.1 adapted to accept continuous action inputs during evaluation?

**Question (2)**: YUME [1] also extends Wan-2.1 [6] for interactive video prediction. Could the authors explain why Wan-2.1 is chosen as the base model instead of YUME?

**Question (3)**: Astra-Bench uses both MegaSaM [7] and human evaluations for “instruction following.” Could the authors clarify how the numerical values of "instruction following" in Tables 1 and 2 are computed?

**Question (4)**: All experiments train on a dataset mixture across five domains. There is no zero-shot evaluation on held-out domains. Does the model generalize to completely new environment unseen during training?

**Question (5)**: Authors claim that increasing the length of history improves temporal consistency but weakens responsiveness (Line 257). However, no supporting quantitative data are provided. Could the authors supply such evidence?

**Typographical Error**: “Eperimental” → “Experimental” (Line 594).

[6] Wan, Team, Ang Wang, Baole Ai, Bin Wen, Chaojie Mao, Chen-Wei Xie, Di Chen et al. "Wan: Open and advanced large-scale video generative models." arXiv preprint arXiv:2503.20314 (2025).

[7] Li, Zhengqi, Richard Tucker, Forrester Cole, Qianqian Wang, Linyi Jin, Vickie Ye, Angjoo Kanazawa, Aleksander Holynski, and Noah Snavely. "MegaSaM: Accurate, fast and robust structure and motion from casual dynamic videos." In Proceedings of the Computer Vision and Pattern Recognition Conference, pp. 10486-10496. 2025.

---

> ### Author Response · Authors · 2025-11-20
> **Official Response to Reviewer toMU (Part 1)**
>
> We sincerely thank Reviewer toMU for the valuable insights and constructive feedback. Below, we address your concerns and questions in detail.
> > ### **About definition and formulation of action signals**
>
> **[Reply]:** Thank you for the insightful comment! We agree that the representation and processing of heterogeneous action signals should be described more clearly. In the revision, we have refined Section 3.4 to more explicitly describe:
> - How each action modality is encoded, including its raw format;
> - How actions are fed into the modality-specific projector;
> - How the router selects the appropriate expert;
>
> These clarifications make the action formulation and integration pathway fully transparent.
>
> > ### **About more comparisons with YUME**
>
> **[Reply]:** We thank the reviewer for pointing this out. While YUME was compared to Astra in our experiments, we agree that it was not sufficiently discussed in the Related Work. In the revision, we have explicitly included YUME in Section 2, highlighting the differences. Specifically, Astra introduces an action-aware adapter, noise-as-mask memory, and a MoAE to handle heterogeneous action modalities and interactions, whereas YUME relies on Masked Video Diffusion Transformer and camera motion quantization.
>
> > ### **About ablation on MoAE**
>
> **[Reply]:** We appreciate the reviewer’s suggestion regarding ablations for MoAE. To validate its contribution, we conducted an additional ablation comparing the full MoAE against a simpler shared-action encoder variant without the routing network. Results show that MoAE consistently improves video quality and action responsiveness. These findings confirm that MoAE meaningfully enhances performance beyond a naive shared representation. We have included this ablation in the revised paper (Table 2 and Section 4.3):
> | Method               | Instruction Following ↑ | Subject Consistency ↑ | Background Consistency ↑ | Motion Smoothness ↑ | Aesthetic Quality ↑ | Imaging Quality ↑ |
> |----|----|---|---|--|---|---|
> | w/o MoAE   | 0.651 | 0.930 | 0.941  | 0.975   | 0.520  | 0.727 |
> | **Astra (Ours)**  | **0.669**  | **0.939**  | **0.945**  | **0.989**| **0.531** | **0.747** |
>
>
> > ### **About action input of Wan-2.1**
>
> **[Reply]:** We use the Wan-2.1-I2V model as the baseline and provide actions via **textual prompts**, such as “turn left” or “move right,” following the evaluation protocol used in YUME.
>
> > ### **About base model choice**
>
> **[Reply]:** We chose Wan-2.1 as the backbone instead of YUME for two main reasons:
> - Our framework differs substantially from YUME in terms of architecture and training strategy. Since our goal is to evaluate our design choices on a general video model and explore how to adapt it into an interactive world model, Wan-2.1 provides a more appropriate base.
> - YUME relies on a 13B-parameter version of Wan-2.1, which is beyond our computation budget. Using the smaller Wan-2.1 backbone allows us to efficiently conduct experiments while focusing on the contributions of our proposed modules.
>
> > ### **About out-of-domain results**
>
> **[Reply]:** We thank the reviewer for the suggestion. We have included results on unseen scenes in **Figure B** of the appendix, including indoor environments, Minecraft worlds, and animation-style videos. Astra produces consistent, coherent, and action-conditioned rollouts across all these out-of-distribution scenarios.
>
> > ### **About evidence of visual inertia**
>
> **[Reply]:** Thanks for your advice. We have provided curves in **Figure E** showing instruction-following performance and video quality as a function of history video length. The results clearly show that as the history length increases, video quality improves, but the action-following score drops sharply, providing direct evidence of visual inertia. We have included these discussions in **Section D**.
>
> > ### **About typos**
>
> **[Reply]:** We thank the reviewer for the careful reading and we have corrected these typos.

---

> ### Author Response · Authors · 2025-11-25
> **Official Response to Reviewer toMU (Part 2)**
>
> > ### **About more quantitative comparison**
>
> **[Reply]:** We thank the reviewer for the comment. While we initially focused on comparisons with the most closely related methods following prior works such as YUME and MatrixGame, we agree that including additional baselines strengthens the evaluation. In the revised paper, we have added quantitative comparisons in **Tables B and C** with **NVM [4]** mentioned by the reviewer. (Vid2World [3] and Genie [5] are **not open-sourced**, and WorldVLA [2] did **not provide video generation code**). We also provide the qualitative comparisons in **Figure D**.
>
> > ### **About instruction following score**
>
> **[Reply]:** We follow YUME to use human evaluation and VBench metrics to assess the performance of our model. For instruction following, we rely purely on **human evaluation**. Specifically, we recruit 20 users to inspect each generated sequence together with the corresponding action command. The instruction-following score is computed as the ratio of users who agree that the generated motion faithfully reflects the specified action direction. For a more objective evaluation, we follow prior works and measure how closely the camera motion in the generated video matches the ground-truth trajectory (**RotErr** and **TransErr**), using MegaSaM to estimate camera poses. We have added this details in **Section A.4** and **Table B**.
>
> We include the newly added Table B below:
> | Method             | RotErr ↓ | TransErr ↓ | Instruction Following ↑ | Imaging Quality ↑ |
> |-------------------|-----------|------------|------------------------|-----------------|
> | Wan-2.1    | 2.96      | 7.37       | 0.061                  | 0.691           |
> | YUME        | 2.20      | 5.80       | 0.268                  | 0.741           |
> | MatrixGame  | 2.25      | 5.63       | 0.652                  | **0.748**       |
> | NWM [4]  | 2.47      | 6.13       | 0.311                  | 0.635           |
> | Astra (ours)          | **1.23**  | **4.86**   | **0.669**              | 0.747           |

---

### Author Response · Authors · 2025-11-20

We sincerely thank all reviewers for their constructive feedback and valuable suggestions. We are encouraged by the positive reception of our work, with all reviewers recognizing our designs—including the ACT-Adapter, noise-as-mask mechanism, and MoAE—as sound, practical, and technically well-motivated. Reviewer toMU described our design choices as "a simple, elegant and original mechanism to amplify action effects without heavy architectural changes", while Reviewer qwc8 highlighted its "innovative" nature. All reviewers also noted the strength of Astra’s general applicability across multiple domains.

We have carefully addressed the raised concerns and clarified potential confusions by **incorporating corresponding modifications into our paper (with revisions highlighted in blue)**.

---

### Meta-Review · Area_Chair_Bj2M · 2026-01-06

**Summary:**

All four reviewers recognized the technical contributions of Astra, particularly praising the ACT-Adapter design, noise-augmented history memory mechanism, and Mixture of Action Experts (MoAE) as elegant and well-motivated solutions for interactive world modeling. The reviewers acknowledged the paper's strength in achieving general-purpose action-conditioned video generation across diverse domains. Several concerns were raised: insufficient discussion of related work (particularly YUME), limited ablation studies (especially for MoAE), unclear action signal formulation, lack of quantitative comparisons with existing methods, concerns about evaluation benchmark scale, questions about the fairness of comparisons between domain-specific and general models, and requests for out-of-domain generalization evidence.

**Reviewer Concerns:**

The authors provided a rebuttal with revisions including new ablation studies, expanded comparisons, and additional experimental results. Key concerns that were effectively addressed include: the MoAE ablation was added (Table 2) demonstrating clear improvements; quantitative comparisons with NVM and detailed metrics were included; YUME was properly discussed in the related work section; action signal formulation was clarified in Section 3.4; out-of-domain results were provided showing generalization to Minecraft, indoor scenes, and multi-agent scenarios; evidence for visual inertia was added; and the evaluation was expanded to 100 scenes from CityWalker. While some concerns about the scope of "general" modeling and fine-grained manipulation quality remain as acknowledged future work, the core technical contributions and experimental validation were strengthened considerably.

**Reviewer Scores:**

Reviewer toMU (initial score: 6) would likely maintain or increase to 6-7 given the added comparisons and ablations. Reviewer AW9Q (initial score: 4) would likely increase to 5-6 following comprehensive responses and parameter analysis. Reviewer ynyx (initial score: 6) would likely maintain their score of 6. Reviewer qwc8 (initial score: 4) would likely increase to 5-6 with the expanded evaluation addressing their main concerns.

---

### Decision · Program_Chairs · 2026-01-26

Accept (Poster)